# The Short-Term Effect of Dynamic Tape versus the Low-Dye Taping Technique in Plantar Fasciitis: A Randomized Clinical Trial

**DOI:** 10.3390/ijerph192416536

**Published:** 2022-12-09

**Authors:** Aurora Castro-Méndez, Inmaculada C. Palomo-Toucedo, Manuel Pabón-Carrasco, Mercedes Ortiz-Romero, Lourdes Mª Fernández-Seguín

**Affiliations:** 1Podiatry Department, University of Seville, 41009 Seville, Spain; 2Nursering Department of La Cruz Roja, University of Seville, 41009 Seville, Spain; 3Physiotherapy Department, University of Seville, 41009 Seville, Spain; 4Institute of Biomedicine of Seville, 41013 Seville, Spain

**Keywords:** fasciitis plantar, orthotic tape, pain measurement, foot

## Abstract

Background: Plantar fasciitis is a painful disorder that affects the plantar fascia of the foot, with a multifactorial aetiology. Dorsal flexion deficiency in the ankle is a risk factor for it. The provisional use of taping is described as part of conservative treatment. Dynamic Tape^®^ is a type of tape that, adhered to muscles, allows for potential elastic energy to accumulate and dissipate later, optimizing its function. Therefore, it can offer immediate benefits while the patient awaits definitive treatment depending on the cause. Objective: To verify the effectiveness of Dynamic Tape^®^ and the low-dye taping technique on pain intensity, ankle range of motion, and foot posture index. Method: A randomised, double-blind clinical trial was conducted. A total of 57 subjects from the Clinical Podiatry Area of the University of Seville (Spain), clinically diagnosed with plantar fasciitis, were randomized into two groups. For one week, in the gastrocnemius–Achilles–plantar system, one group was treated with Dynamic Tape^®^ and another group with low-dye taping. Pain, degrees of movement of dorsal flexion, and the foot posture index were measured in both groups before the intervention and one week after the intervention. A repeated-measures analysis of variance (ANOVA) was used to explore the differences between groups. Results: Significant differences in the decrease in pain intensity using Dynamic Tape^®^ were found when comparing the treatments (*p* = 0.015) and the foot posture index was more normal in low-dye taping (*p* < 0.001). In both cases, the treatment showed similar behaviour with respect to the dorsal flexion ankle movement. Conclusion: The effectiveness of Dynamic Tape^®^, compared to that of the low-dye taping, has a major benefit with regard to pain intensity from fasciitis, although it does not produce changes in the ankle range of motion and foot posture index. Consequently, Dynamic Tape^®^ can be considered a taping technique with beneficial effects on pain intensity in the provisional approach to plantar fasciitis.

## 1. Introduction

Plantar fasciitis (PF) is defined as a painful disorder of the plantar fascia of the foot and produces degenerative changes in its structure [1,2,3,4,5]. Its aetiology is multifactorial and its prevalence may affect up to 10% of the population [1,2]. Excessive stress of this structure is associated with risk factors such as obesity, physical activity, limited ankle dorsal flexion movement, or an anormal foot posture index (usually associated with the contracture of the flexor muscles of the foot) [5,6]. The fascia is integrated into the gastrocnemius–Achilles–plantar system (GAPS), and these structures are decisive with respect to their functional connection to the windlass mechanism. Its functionality could be affected in situations of decreased elasticity of the gastrocnemius muscles [1,2].

The conservative therapeutic approach of PF is typically effective in 90% of cases. These include foot orthoses, nocturnal splinting, pharmacological treatments, infiltrations, gastrocnemius and soleus muscle stretching, physical therapy, or taping [2,6,7]. Different types of taping techniques are described as a provisional treatment of PF, usually applied in short periods (7 to 10 days) until the definitive treatment, such as for example foot orthoses. The low-dye taping (LDT) technique, developed in 1936, is described as the most used in the control of foot pronation, considered effective and commonly used in PF due to its control of the foot posture and foot arch, thus relaxing the fascia [6,7,8,9,10,11,12].

Dynamic Tape^®^ (DT) was designed by a physiotherapist, Ryan Kendrick, in 2010. Its composition of nylon and lycra provides great elasticity (200%) in all directions and resistance. Its application favours a boomerang effect, allowing the storage of potential elastic energy during concentric muscle contraction, which in turn becomes kinetic energy during eccentric contraction. Therefore, it reduces loads, helps mobility, decreases muscle tension, and stores and recovers energy, collaborating in the facilitation of muscle function [13,14,15,16,17].

We consider that DT applied to GAPS can optimize its function, increasing the kinetic energy necessary for walking and reducing the muscular demand on GAPS. During the contact phase of the gait cycle, there is an elongation of GAPS, where the triceps sural muscle structure generates elastic energy, which, later, in the propulsive phase, would be used to reduce the tension of the plantar fascia and, consequently, its pain [2,7,13,15,18].

Therefore, the aim of this study is to evaluate the short-term efficacy of DT in a group of subjects versus another group treated with LDT, both with PF. PF pain, range of movement of the ankle joint (with the knee in extension and in flexion), and foot posture index were compared between both groups after the short-term application of taping, before and one week after regular use.

## 2. Materials and Method

### 2.1. Design

The study design was a parallel clinical trial with a double-blind technique. This trial was registered in ClinicalTrialGov: NCT04022070.

### 2.2. Participants

The total sample was obtained from patients attending the Clinical Podiatry Area (ACP) of the University of Seville and who consulted the clinic about PF pain. The initial number of subjects was 65, but the study was completed with a total of 57 participants (Figure 1).

Patients who voluntarily participated in the investigation signed a consent form prior to participation. The principles set out in the Declaration of Helsinki and all its agreements [19] were complied with and the CONSORT guidelines were followed [20].

The inclusion criteria were subjects ≥18 years old consulting the ACP for bilateral PF (diagnosis based on clinical guidelines linked to the International Classification of Function, Disability and Health of the Orthopaedic Section of the American Physiotherapy Association) [21] and previous symptomatology of at least 4 weeks. Exclusion criteria: pain from previous trauma or injuries, current PF treatment, pregnancy, a rheumatic or autoimmune disease. A minimum 4-week period of evolution was established as recommended by previous research [22].

### 2.3. Randomization and Blinding

An assistant unknown to the next phases of the study randomly assigned a number to each participating subject. In a second moment, the sequence of numbers obtained was randomized into two groups: an LDT group and a DT group. This assistant was the person in charge of giving appointments for each treatment according to the assigned group. Random allocation software 2.0 was used for this process [23]. The intervention in each group was made by two podiatrists specialized in tape with 15 years of experience (one for each group). The investigator in charge of collecting the study variables, or main researcher, was unaware of the treatment allocation group at all times.

### 2.4. Measurements

The demographic variables were age, weight, height, BMI, and the laterality of the member analysed (manual predominance) and were evaluated at the beginning of study. The independent variables were the treatments (DT or LDT) and the dependent variables were VAS to quantify pain (visual analogue scale, 100 mm; 0 mm: no pain and 100 mm: maximal pain) [24], ankle dorsal flexion quantified by the usual technique (two-branch goniometer, in degrees) with the knee extended (DF_EXT) and flexed (DF_FLEX) (Silfverskiöld Test) [5], and the six foot posture index items (FPI) [25]. One group was treated with LDT tape, while another was treated with a DT-type tape, keeping it, in both cases, for 1 week (same use in other investigations) [2,11]. The initial assessment of the variables coincided with the beginning of the study, before the taping (pre-intervention). The second assessment was done after a week of use, when the subject went back to the ACP; the taping was removed, and those variables were re-evaluated (post-intervention).

### 2.5. Study Protocol

The main researcher (a podiatrist also with 15 years of experience) performed a clinical examination to confirm a differential diagnosis of PF from other pathologies [5]. Clinical diagnosis of PF was made according to the Clinical Guidelines linked to the International Classification of Function, Disability and Health of the Orthopaedic Section of the American Physiotherapy Association [21]. The protocol was performed to diagnose PF based on the characteristic signs of scientific evidence:-Windlass mechanism assessment.-Characteristic pain in the proximal medial calcaneal region.-Increased pain intensity after rest or exercise.-Test negative for tarsal tunnel syndrome.-FPI alteration.-Increase in BMI.

For the taping application, first, participants had the hair on their skin removed and the skin cleaned with alcohol to increase adhesion. Subsequently, the area was sprayed with a special adhesive spray to increase its adhesion and avoid allergic reactions to the tape.

The direct Achilles technique described by McNeill and Pedersen was used to apply the DT. A five-centimetre-wide DT was applied on the patient in a prone position, from the metatarsophalangeal joints, over the Achilles tendon to the proximal 2/3 of the leg, as described in the technique. A 200% stretch nylon webbing was used but no stretching in the tape was necessary because of the shortened start position [13]. LDT consists of a rigid cotton tape of high traction resistance [12]. The technique described by Osborne and Allison was used. In this technique, the tape strips adhere with the intention of positioning the foot in the most neutral position possible, thus reducing the tension of the fascia (Figure 2). Transverse tape strips were combined with cross strips from the metatarsal and around the heel.

### 2.6. Statistical Analysis

The sample size was calculated with the software Gpower 3.1.0 (Franz Faul, Universität Kiel, Germany); the test family was selected: F Test; statistical test: ANOVA fixed effects. Omnibus, one-way and type of power analysis was a priori. The power was 0.95, with an alpha error of 0.05 and a size effect of 0.5. A total sample size of 54 subjects was necessary (57 participants finally adhered to the study).

SPSS^®^ 24.0 software was used. A prior descriptive analysis was performed of qualitative variables (relative frequencies) and quantitative variables (central tendency and dispersion measures). They were identified through an exploratory analysis prior to the distribution of variables (Kolmogorov–Smirnov test). The U Mann–Whitney test was applied for independent groups (a test to check the homogeneity of the variables). A repeated-measures analysis of variance (ANOVA) was used to compare the differences in the variables’ changes one week after intervention between groups. The level of significance was set to *p* < 0.05.

## 3. Results

A total of 57 adults (28 women and 29 men) with a mean age of 41.7 ± 8.9 years, weight of 75.5 ± 11.8 kg, height of 1.67 ± 0.08 mm, and BMI of 25.25 ± 3.93 was recruited (Table 1). At baseline, there were no differences between the clinical and demographic features of the participants recruited (*p* > 0.05). Table 2 shows pre- and post-treatment data for the dependent variables for each group.

Table 3 shows the changes in the variables one week after the placement of the tape and the comparison between groups. In the DT group, ANOVA showed a significant decrease in changes in VAS (*p* = 0.015) with an effect value very close to the high value (η^2^ =0.10; value high effect η^2^ = 0.14). In the LDT group, the changes in FPPI were significant (*p* < 0.001) with a high effect size. The changes experienced in the FD_EXT and FD_FLEX variables were not significant when comparing one group with another. Calculated on the VAS variable, the sample effect size was 0.665 and the sample power was 0.99.

## 4. Discussion

The main aim of this study was to compare the effect of DT, after one week of regular use, with LDT as the gold standard of taping [11]. The results of this investigation suggest that taping treatment using DT has a major beneficial effect on reducing PF pain compared to the LDT technique. No significant changes between the two groups are evident when comparing the range of motion of the DF.

There is little scientific evidence that examines the effect of DT on pain and range of motion of ankle DF and foot posture index in subjects with plantar fasciitis. It is important to explain that taping is a temporary treatment, and its function is not based on modifying the biomechanical parameters of the foot; in this case, it is intended to improve PF pain. The main purpose of PF taping is to reduce tension on the plantar fascia to relieve pain while awaiting final treatment [2,6,11].

As discussed above, taping applied for PF reduces pain in the short term, is a temporary treatment, and its effect is based on reducing tension [8,9,10,11,12]. During the mid-stance phase of the gait cycle, there is a moment of increased tension on the fascia; all the body weight falls on the reference foot and causes the inner arch to descend and the foot to pronate, increasing pressure. For an active propulsive gait and the foot to function as a rigid lever, a correct windlass mechanism (GAPS integrated) is necessary to transmit the weight forward (in the anterior direction) and the foot can act as a rigid lever for a correct propulsion (toe-off) [1,3,5,6,26,27,28].

In subjects with plantar fasciitis, the pain is caused by excessive tension in the plantar fascia [29]. Helping normal biomechanical movement is one of the properties of the DT [13]. Thus, we could think that the improvement of pain could be justified by the help that DT provides to the windlass mechanism in the propulsive phase. Electromyographic studies show that elastic tapes inhibit muscle activity [30]. Therefore, as it takes part in generating energy for the posterior sural muscle unit, pressure there decreases, and, consequently, that of the plantar fascia due to its integration with GAPS [15]. These data are consistent with the benefits obtained in previous research [31,32,33].

Robinson et al. [31] conducted a cross-clinical trial of two interventions with simple randomization in 50 women affected by trochanteritis. These interventions consisted of (1) DT application and (2) a cross-control tape. A three-dimensional analysis of the gait was performed before and after the application of the treatments. Pre- and post-intervention measurements were performed; kinetic variables corresponding to the adduction movement of the hip joint, its internal rotation, and the obliquity of the pelvis were recorded. The results showed significant differences in pain and kinetic variables collected in both interventions, particularly in hip adduction and pain. The results were argued by proving that pressure was reduced in the tendon with the help of the tape’s muscle-level function, thus reducing pain. Based on the results, DT was shown to modify some kinematic gait patterns.

Following the same line of thought regarding PF pain improvement and the use of DT, and linking, in turn, with our findings, De la Cruz et al. assessed the immediate effect of DT on the degrees of external tibial torsion of a footballer with painful symptoms. They applied the tape and compared the values of the variable thigh-foot angle, pain, and electromyography prior to and after 10 days of use. The results showed a decrease in the thigh-foot angle of the lower right limb and a 55% pain reduction. In turn, the tension on the muscle was modified; on one hand, the muscles with initial high tension experienced a reduction in muscle activation, and on the other, muscles that were weakened in the situation prior to taping increased their muscle activation. They concluded that the dynamic tape could be considered a very effective complementary therapeutic tool to reduce external tibial torsion and pain [14]. A similar conclusion was reached in this study in which both pain and tension in the fascia changed, influencing PF development.

Twenty elite volleyball players with a history of lateral ankle sprain were studied without a control group; ankle instability was assessed using the Cumberland tool (CAIT), a balance test with and without the dynamic tape. The results showed better records in all cases [32]. Evans analysed the effect of this tape on the width of the foot and ND on athletes after a resistance jump against a control group. It was concluded that DT controlled the fall of the inner longitudinal and lateral arches of the foot [34].

With regard to the scientific evidence on LDT application, it upholds its kinetic and kinematic movement with control of the effect of the pronating force, thus reducing the pain and tension of the plantar fascia during the gait [7,11,27,28]. This agrees with our results where LDT assessed significant changes in FPPI compared to DT. In this study, DT has been used as a direct Achilles technique and it is not intended to modify the posture of the foot. This technique is specifically used to relieve plantar fasciitis pain by working with the gastrocnemius–soleus–Achilles tendon (GAPS) in withstanding dorsal flexion and transitioning to plantar flexion to reduce the burden on the Achilles tendon [13].

Some limitations of this study should be mentioned. Although taping is generally considered a provisional therapy, the trial has assessed short-term changes (1 week), so the duration of the changes over time is not known. On the other hand, these data could not be applied to the general population because the sample of this study consists of a small number of randomly selected participants. Thereover, the subjects were diagnosed according to the clinical guidelines linked to the International Classification of Function, Disability and Health of the Orthopaedic Section of the American Physiotherapy Association but a functional study of the fascia was not performed. For that, future studies are needed.

## 5. Conclusions

Dynamic Tape^®^ (DT) in comparison with LDT has shown beneficial effects regarding PF treatment decreasing the pain of patients. However, it does not offer a biomechanical correction in the posture of the foot as LDT. Both treatments are regarded as appropriate provisional treatments of PF. However, more detailed studies are needed to assess the benefits of Dynamic Tape^®^.

## Figures and Tables

**Figure 1 ijerph-19-16536-f001:**
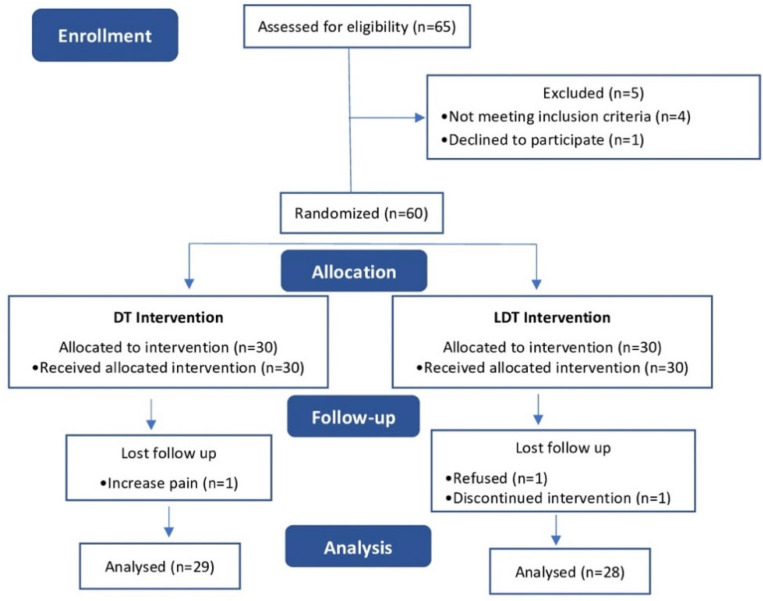
CONSORT flow chart.

**Figure 2 ijerph-19-16536-f002:**
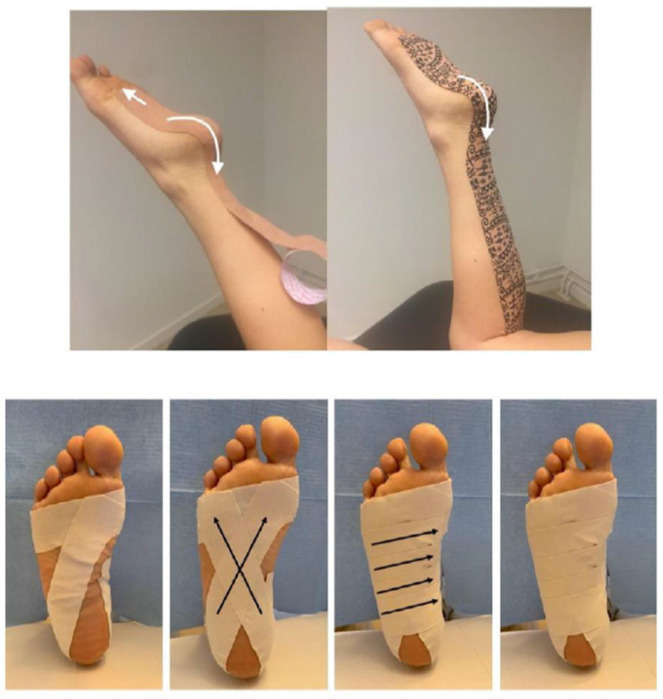
Dynamic tape technique and low-dye taping technique.

**Table 1 ijerph-19-16536-t001:** Demographic data of groups.

	DT Group (*n* = 29)	LDT Group (*n* = 28)
Age	39.59 ± 9.55	44.43 ± 7.06
Gender male	15 (51.7%)	14 (50.0%)
Gender female	14 (48.3%)	14 (50%)
Weight	75.21 ± 15.23	75.71 ± 6.97
Height	169.28 ± 10.65	165.29 ± 4.52
IMC	26.11 ± 4.07	24.38 ± 3.64

DT: Dynamic Tape^®^; LDT: low-dye taping; IMC: body mass index.

**Table 2 ijerph-19-16536-t002:** Descriptive statistics of groups.

		Mean SD	95 IC	*p* ^a^
VAS	PRE			
	DT group	7.55 ± 1.38	7.03–8.08	
	LDT group	6.64 ± 2.87	5.53–7.76	0.20
	POST			
	DT group	5.35 ± 1.15	5.11–5.99	
	LDT group	5.54 ± 3.30	4.26–6.81	
DF_EXT	PRE			
	DT group	7.50 ± 5.05	5.06–9.43	
	LDT group	8.32 ± 1.65	7.67–8.96	0.73
	POST			
	DT group	8.45 ± 4.90	6.57–10.32	
	LDT group	8.56 ± 1.60	7.91–9.15	
DF_FLEX	PRE			
	DT group	15.17 ± 6.31	12.77–17.57	
	LDT group	15.42 ± 4.57	13.65–17.20	0.78
	POST			
	DT group	15.69 ± 6.20	13.32–18.05	
	LDT group	16.36 ± 4.50	14.61–18.10	
FPI	PRE			
	DT group	2.90 ± 3.68	1.60–4.30	
	LDT group	3.96 ± 3.70	2.53–5.40	0.19
	POST			
	DT group	2.93 ± 3.67	1.53–4.33	
	LDT group	3.50 ± 3.99	1.95–5.05	

DT: Dynamic Tape^®^; LDT: low-dye taping; VAS: visual analogic scale; DF_EXT: ankle dorsal flexion with the knee extended; DF_FLEX: ankle dorsal flexion with the knee flexed; FPI: foot posture index. ^a^ Mann–Whitney U test.

**Table 3 ijerph-19-16536-t003:** Changes in variables from baseline to one week after tape colocation. Comparison between groups. Mean ± standard deviation.

Changes (POST-PRE Values)	Group	Mean SD	95 IC	*p* ^b^	η^2^	F
VAS	DT	−2.05 ± 0.96	(−2.37)–(−1.63)	0.015 *	0.10	6.33
	LDT	−1.10 ± 1.64	(−1.74)–(−0.47)			
DF_EXT	DT	0.93 ± 2.17	0.10–1.7	0.09	0.05	2.86
	LDT	0.21 ± 0.56	(<−0.01)–0.43			
DF_FLEX	DT	0.52 ± 2.89	(−0.58)–1.6	0.47	0.01	0.51
	LDT	0.93 ± 0.94	0.56–1.2			
FPI	DT	0.034 ± 0.32	(−0.08)–0.15	<0.001 *	0.20	14
	LDT	−0.47 ± 0.64	(−0.71)–(−0.22)			

DT: Dynamic Tape^®^; LDT: low-dye taping; VAS: visual analogic scale; DF_EXT: ankle dorsal flexion with the knee extended; DF_FLEX: ankle dorsal flexion with the knee flexed; FPI: foot posture index. η^2^: effect size. F: variation between sample means /variation within the samples. ^b^ ANOVA test. * *p* < 0.05.

## Data Availability

The data associated with the paper are not publicly available but are available from the corresponding author upon reasonable request.

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
