# Peer review of "The Short-Term Effect of Dynamic Tape versus the Low-Dye Taping Technique in Plantar Fasciitis: A Randomized Clinical Trial"

_ijerph, 2022, doi:10.3390/ijerph192416536_

Round 1

Reviewer 1 Report

It was a great pleasure to evaluate your manuscript. Some corrections are necessary for better presentation of the manuscript.

 Abstract: Background: the defining sentence should be better described, what is the importance of this study? Objective: I suggest adding the variables that will be analyzed in the study. Example: To verify the effectiveness of Dynamic Tape® and the Low Dye-Taping technique on pain intensity, ankle range of motion and foot posture index. Method: describe the study design. It describes which statistical test was used for comparison between groups and between times. Results: I suggest adding “pain intensity” to the pain variable (visual analogue scale measured pain intensity). Has the statistical difference for pain intensity increased or decreased? Did the foot posture index variable show an increase or decrease in this index? Conclusion: I believe it needs change; conclusion must be in line with the objectives.

 Introduction

It presents contextualization. It presents a study on the treatment approach. However, what does the study bring as an innovation? Example: treatment time? How to place the taping? Assessment instruments? At the end of the introduction, I suggest adding the hypothesis of the study.

 Methodology

Line 108 – The pain intensity outcome (VAS) needs to have more details in the evaluation form. Example: 0 corresponds to improvement or worsening? 100mm corresponds to improvement or worsening? How was the question asked? Was this variable performed by a trained researcher? Was this researcher who evaluated the same one who treated?

The range of motion outcome of ankle pain flexion. What position was the volunteer in during this assessment? In which region was the goniometer placed?

The six foot posture index outcome – How is it done? How is this review scored?

 Results

Present the characteristics of the sample separated by group and their statistics (age, body mass, height, body mass index). If you have the time when the volunteers had pain, it would be interesting to present.

Table 1. I suggest changing the table format. Check the articles published in the magazine that present two groups. It's not presentable. I suggest removing the maximum and minimum and putting the 95% confidence interval. Describe the p-value between times in the table. The letter “p” in the confidence interval must be lowercase. Describe in the caption the statistical test used.

 Table 2. I suggest changing the table format. I suggest adding the 95% confidence interval. Add the meaning of F and n2 in the table caption and insert *p<0.05. Describe in the caption the statistical test used. Add the “ * ” where there is a difference. Add the mean of the difference and the 95% confidence interval of the mean for the variables that had a difference. Add effect size and sample power (post hoc).

 Discussion

Line 264 the researcher points out that the sample was for convenience, I did not understand why in the methodology it was highlighted that it was a randomized clinical trial. What is right?

Author Response

Dear Reviewer 1

Ref.: Submission ID: ijerph-2082176

Manuscript entitled: " The Short-Term Effect Of The Dynamic Tape Versus Low Dye-Taping Technique In The Plantar Fasciitis: A Randomized Clinical Trial."

Dear Reviewer:

Thank you for giving us the opportunity to resubmit our work. We have revised the manuscript according to the suggestions and comments made by you. We made a point-by-point response to the comments and edited our manuscript accordingly. Changes are marked using the “Track Changes” of word in the order suggested by the reviewers. We thank you in advance for taking time to consider our manuscript and eagerly await your response.

Yours faithfully,

Dr. Lourdes María Fernández-Seguín

Reviewer 2 Report

In this paper (ijerph-2082176), the authors evaluated the short-term efficacy of DT in a group of subjects versus another group treated with LDT. The results are useful and the topic can attract a wide range of readerships. But there are some problems in the writing, presentation, and discussion of the results. As such, some revisions are needed before possible publication. My specific comments are as follows:

1.      Double blind experiment is recommended.

2.      During the treatment, whether some personal behaviors or activities of the patient have a certain impact on the efficacy of each group.

3.      It is recommended to evaluate the safety of adhesive tape, such as whether the patient has skin allergy after one week of use.

4.      English writing needs to be checked.  

5.      Three-line table is recommended. Check the font format (font size is not uniform).

6.      Check the reference format of the journal.

Author Response

Dear Reviewer:

Thank you for giving us the opportunity to resubmit our work. We have revised the manuscript according to the suggestions and comments made by you. We made a point-by-point response to the comments and edited our manuscript accordingly. Changes are marked using the “Track Changes” of word in the order suggested by the reviewers. We thank you in advance for taking time to consider our manuscript and eagerly await your response.

Yours faithfully,

Dr. Lourdes María Fernández-Seguín

In this paper (ijerph-2082176), the authors evaluated the short-term efficacy of DT in a group of subjects versus another group treated with LDT. The results are useful and the topic can attract a wide range of readerships. But there are some problems in the writing, presentation, and discussion of the results. As such, some revisions are needed before possible publication. My specific comments are as follows:

  1. Double blind experiment is recommended.

      We changed single for double as both therapist and patient did not know the intention of the treatment applied in each group.

  1. During the treatment, whether some personal behaviors or activities of the patient have a certain impact on the efficacy of each group.

      We did not detect any behavior or activities of the patient that would influence the effectiveness of the applied treatment. The tape was kept for one week in all cases in both groups.

  1. It is recommended to evaluate the safety of adhesive tape, such as whether the patient has skin allergy after one week of use.

      None of the 57 patients had allergies to the applied treatment.

  1. English writing needs to be checked.  

      We have checked the english writing.

  1. Three-line table is recommended. Check the font format (font size is not uniform).

      We have changed the tables.

  1. Check the reference format of the journal.

      We have checked all reference.

Round 2

Reviewer 2 Report

I carefully checked the response and revised manuscript. Concerns of  reviewer  have been considered and addressed properly, and publication is recommended.